# 4-(2,5-Dimethyl-1*H*-pyrrol-1-yl)-*N*-(2,5-dioxopyrrolidin-1-yl) benzamide improves monoclonal antibody production in a Chinese hamster ovary cell culture

Yuichi Aki[1,2]☯*, Yuta Katsumata[1‡], Hirofumi Kakihara[1‡], Koichi Nonaka[1‡], Kenshu Fujiwara[2☯]

**1** Biologics Division, Biologics Technology Research Laboratories, Daiichi Sankyo Co., Ltd., Chiyoda-machi, Gunma, Japan, **2** Department of Life Science, Akita University, Tegata Gakuen-machi, Akita, Japan

☯ These authors contributed equally to this work.
‡ These authors also contributed equally to this work.
* aki.yuichi.ha@daiichisankyo.co.jp

## Abstract

There is a continuous demand to improve monoclonal antibody production for medication supply and medical cost reduction. For over 20 years, recombinant Chinese hamster ovary cells have been used as a host in monoclonal antibody production due to robustness, high productivity and ability to produce proteins with ideal glycans. Chemical compounds, such as dimethyl sulfoxide, lithium chloride, and butyric acid, have been shown to improve monoclonal antibody production in mammalian cell cultures. In this study, we aimed to discover new chemical compounds that can improve cell-specific antibody production in recombinant Chinese hamster ovary cells. Out of the 23,227 chemicals screened in this study, 4-(2,5-dimethyl-1*H*-pyrrol-1-yl)-*N*-(2,5-dioxopyrrolidin-1-yl) benzamide was found to increase monoclonal antibody production. The compound suppressed cell growth and increased both cell-specific glucose uptake rate and the amount of intracellular adenosine triphosphate during monoclonal antibody production. In addition, the compound also suppressed the galactosylation on a monoclonal antibody, which is a critical quality attribute of therapeutic monoclonal antibodies. Therefore, the compound might also be used to control the level of the galactosylation for the *N*-linked glycans. Further, the structure-activity relationship study revealed that 2,5-dimethylpyrrole was the most effective partial structure of 4-(2,5-dimethyl-1*H*-pyrrol-1-yl)-*N*-(2,5-dioxopyrrolidin-1-yl) benzamide on monoclonal antibody production. Further structural optimization of 2,5-dimethylpyrrole derivatives could lead to improved production and quality control of monoclonal antibodies.

## Introduction

The use of therapeutic monoclonal antibodies (mAb) is becoming more common in the treatment of cancer (pembrolizumab, nivolumab, trastuzumab, etc.) and autoimmune diseases

**Data Availability Statement:** The chemical compound library used for screening in this study is owned by the Basis for Supporting Innovative

Drug Discovery and Life Science Research (BINDS) program. BINDS has contractually imposed upon us restrictions involving data availability. The contact information of the data access committee and other institutional bodies are as follows: Data access committee Basis for Supporting Innovative Drug Discovery and Life Science Research (BINDS) 1-1-1, Yayoi, Bunkyo-ku, Tokyo 113-8657, Japan E-mail: assist@mail. ecc.u-tokyo.ac.jp Website: https://www.binds.jp/ Other institutional bodies 1. University of Tokyo 7-3-1, Hongo, Bunkyo-ku, Tokyo 113-0033, Japan E-mail: ddiinfo@mol.f.u-tokyo.ac.jp 2. Osaka University 1-6, Yamadaoka, Fukida-shi, Osaka 565-0871, Japan E-mail: yakugaku-syomu@office. osaka-u.ac.jp 3. Tohoku University 6-3, Aoba, Aramaki-aza, Aoba-ku, Sendai-shi, Miyagi 980-8578, Japan E-mail: www@mail.pharm.tohoku.ac. jp There are no restrictions on the other data we studied. These data are included in the paper and Supporting Information files.

**Funding:** 1. The chemical compound library used in this study was provided by the Platform Project for Supporting Drug Discovery and Life Science Research (the Basis for Supporting Innovative Drug Discovery and Life Science Research [BINDS] program) from the Japan Agency for Medical Research and Development (AMED) under grant numbers JP19am0101086, JP19am0101084, and JP19am0101095 (support numbers 1009, 0847, and 1011, respectively). The funders had no role in the study design, data collection, data analysis, decision to publish, or preparation of the manuscript. 2. Daiichi Sankyo Co., Ltd., provided support in the form of salaries for the authors and supporters (YA, YK, HK, KN, AK, and MK), but the company did not have any additional role in the study design, data collection, data analysis, decision to publish, or preparation of the manuscript. The specific roles of these authors are articulated in the Author Contributions section.

**Competing interests:** Daiichi Sankyo Co., Ltd. employs some authors [YA, YK, HK, and KN]. This commercial affiliation does not alter our adherence to all PLOS ONE policies on sharing data and materials.

(adalimumab, infliximab, etc.) [1–7]. Pharmaceutical and biologics companies have strongly focused on increasing mAb supply and reducing manufacturing costs for commercialization and sustainable growth. MAb concentrations of several grams per liter, produced by using recombinant Chinese hamster ovary (rCHO) cells harboring the genes encoding for mAb, has been achieved in fed-batch cultures through effective host-vector system development [8, 9], genetically engineered host breeding [10], custom media development [11, 12], and culture parameter optimization [13, 14].

Efforts have been continued to establish cost-effective manufacturing processes by reducing the use of expensive materials such as cell culture media and column resins. It is also known that adding chemical compounds in cell cultures improves mAb production. Various chemical compounds such as dimethyl sulfoxide (DMSO) [15], lithium chloride [16], butyric acid [17], valeric acid [18], valproic acid [19], phenolic antioxidants [20], thymidine [21], and CDK4/6 inhibitor [22] have the potential to enhance mAb production when added to the cell culture at an adequate concentration to control the intracellular state of rCHO cells. In addition, 3-methyladenine up-regulates the unfolded protein response pathway and improves production [23], and compound 7312, which works as a caspase inhibitor, suppresses apoptosis and improves production [24].

To pursue the possibility of the chemical compound approach, potential synthetic compounds that might enhance mAb production in rCHO cells were gathered from the Basis for Supporting Innovative Drug Discovery and Life Science Research (BINDS) program and evaluated by screening in batch cultures. We discovered that 4-(2,5-dimethyl-1*H*-pyrrol-1-yl)-*N*-(2,5-dioxopyrrolidin-1-yl) benzamide (MPPB), which was developed as an anti-tuberculosis therapeutic compound [25–27], stimulated mAb production in cell cultures. To reveal the characteristics of the compound, we investigated cell culture trends and the metabolism of rCHO cells under MPPB-supplemented conditions. These studies were executed in both batch and fed-batch cultures. The results showed that MPPB improved mAb production while retaining viability and increasing cell-specific productivity by suppressing cell growth and increasing the cell-specific glucose uptake rate and the amount of intracellular adenosine triphosphate (ATP). MPPB not only improved mAb production, but also affected the *N*-glycan profile. The structure-activity relationship of MPPB was evaluated using the compounds corresponding to the partial structures of MPPB as additives in batch cultures and demonstrated that 2,5-dimethylpyrrole enhanced the cell-specific productivity.

## Materials and methods

### Cell line and cell culture media

The tested rCHO cells, which expressed the mAb (Immunoglobulin G1), were generated from CHO-S host cells (Thermo Fisher Scientific, Waltham, MA, USA) by Daiichi Sankyo Co., Ltd. In-house chemically defined media in which L-alanyl-L-glutamine concentration was adjusted to 1 mM was used as expansion and basal media. In-house chemically defined feed media was used in the fed-batch cultures.

### Chemical compound library

The chemical compound library was provided by the University of Tokyo, Osaka University, and Tohoku University, which are members of Basis for Supporting Innovative Drug Discovery and Life Science Research (BINDS) which is a program that supports drug discovery.

## Expansion culture condition

The rCHO cells were inoculated at a target viable cell concentration of $0.3 \times 10^6$ cells/mL in 125-mL or 250-mL Erlenmeyer flasks (Corning, Corning, NY, USA) with expansion medium. The working volume was adjusted to 50 mL in the 125-mL flasks and to 100 mL in the 250-mL flasks. The culture solutions were incubated for 3 to 4 days at 37.0˚C in 5% $CO_2$-enriched air with the shaker set to 120 rpm (20-mm stroke length).

## Screening of the chemical compound library

**Initial screening.** The rCHO cells were inoculated at a target viable cell concentration of $0.3 \times 10^6$ cells/mL in CELLSTAR 96-well suspension culture plates (Greiner Bio-One, Frickenhausen, Germany) with expansion medium (198 μL). Rows 1 and 12 on the 96-well plates were used as controls with DMSO (2 μL). On day 0, 2 μL of each chemical compound dissolved in DMSO was added to each well. The plates were incubated at 37.0˚C with the shaker set to 120 rpm (20-mm stroke length). After 3 days of culture, mAb concentrations were analyzed with Octet QKe (ForteBio, Fremont, CA, USA). The resulting mAb concentrations were converted to arbitrary units according to the following Eq (1).

$$\textbf{Arbitrary unit} = (\textbf{tested mAb concentration} - (\textbf{average mAb concentration of the controls conditions} + \textbf{3 standard deviations})) \qquad (1)$$

The compounds for which the arbitrary unit showed a positive value were selected as candidates for the secondary screening. The validity of this screening protocol is shown in S1 Table.

**Second screening.** In the second screening to evaluate the candidates, rCHO cells were inoculated at a target viable cell concentration of $0.3 \times 10^6$ cells/mL in 50-mL suspension culture flasks (Greiner Bio-One) with basal medium. The working volume was adjusted to 10 mL. The cell culture solutions were incubated for 3 days in 5% $CO_2$-enriched air at 37.0˚C in static culture. All tested compounds were dissolved in DMSO at a concentration of 100 mg/mL, and 10 μL (0.1% v/v) of this solution was added to the culture on day 0.

## Batch culture condition

In the suspension batch cultures, the rCHO cells were inoculated at a target viable cell concentration of $0.3 \times 10^6$ cells/mL in 50-mL suspension culture flasks (Greiner Bio-One) with basal medium. The working volume was adjusted to 10 mL. The rCHO cell seeding flasks were incubated for 3 to 10 days in 5% $CO_2$-enriched air at 37.0˚C with the shaker set to 120 rpm (20-mm stroke length).

## Fed-batch culture condition

The rCHO cells were inoculated at a target viable cell concentration of $0.3 \times 10^6$ cells/mL in 125-mL Erlenmeyer flasks (Corning) with basal medium. The working volume was adjusted to 50 mL. The culture solutions were incubated at 37.0˚C in 5% $CO_2$-enriched air with the shaker set to 120 rpm (20-mm stroke length) until less than 70% viability was achieved. On days 4, 6, and 8, feed medium at 2% v/v against the starting volume was added. On day 0, 50 μL (0.1% v/ v) of 4-(2,5-dimethyl-1*H*-pyrrol-1-yl)-*N*-(2, 5-dioxopyrrolidin-1-yl) benzamide (Abamachem Ltd., Kyiv, Ukraine) dissolved in DMSO at a concentration of 200 mg/mL was added.

## Measurement of viable cell density and viability

Vi-CELL XR (Beckman Coulter, CA, USA) was used to evaluate viable cell density (VCD) and viability according to the manufacturer's instructions. All testing samples were diluted twice except for on days 0 to 4.

## Metabolite analysis

Glucose and lactate concentrations in the cultures were analyzed by BioProfile FLEX2 (Nova Biomedical, Waltham, MA, USA) according to the manufacturer's protocol.

## Concentration of mAb

Cell culture solution was filtrated with a 0.2-μm filter. The mAb concentration in the residual solution was measured by high-performance liquid chromatography (HPLC) with a Protein A affinity chromatography column, which is a PA ImmunoDetection Sensor Cartridge, with an ID of 2.1 × 30 mm (Applied Biosystems, Bedford, MA, USA).

## Measurement of intracellular ATP

An intracellular ATP assay kit (v2, Toyo B-Net, Tokyo, Japan) was used to quantitate ATP amounts in rCHO cells. Each cell culture solution (100 μL) was diluted with phosphate-buffered saline (1 mL) and then centrifuged (450 g, 5 min), and the supernatant was discarded (1 mL). The resulting cell pellet was suspended in phosphate-buffered saline (1 mL) and then centrifuged (450 g, 5 min), and the supernatant was again discarded (1 mL). The cell pellet was suspended in the ATP extraction reagent from the kit (0.5 mL) and incubated for 5 minutes to extract ATP from the cells. The suspension (10 μL) was mixed with luciferase luminescence assay reagents (100 μL), and measurements were obtained by using the Infinite M Plex plate reader (Tecan, Mannedorf, Switzerland). The resulting ATP concentration was converted to the amount of intracellular ATP per cell.

## Calculation of cell-specific productivity, cell-specific glucose uptake rates, and cell-specific lactate production rates

Cell-specific productivity (pg/cell/day), cell-specific glucose uptake rates (pmol/cell/day), and cell-specific lactate production rates (pmol/cell/day) were calculated as the slopes of mAb concentration (mg/L), consumed glucose concentration (mM), and lactate concentration (mM), respectively, to integral viable cell concentration ((cells·day)/mL), according to previous reports [28, 29].

## *N*-linked glycan analysis

The samples purified with the spin column-based antibody purification kit (Cosmo Bio, Tokyo, Japan) according to the manufacturer's protocol were labeled with 2-AB using the EZGlyco mAb-N kit (Sumitomo Bakelite, Tokyo, Japan) for *N*-linked glycan analysis, which was performed under gradient conditions of 50 mM ammonium formate buffer (pH 4.4) on HPLC using a XBridge BEH Amide XP Column (2.5 μm, ID 4.6 × 150 mm, Waters Corporation, Milford, MA, USA) at 60°C. The detected wavelength was 420 nm, and the flow rate was changed from 0.4 to 1.0 mL/min under gradient conditions.

## Statistical analysis

At least 3 batches were executed for statistical analysis. The mean ± standard deviation (SD) and P-value were calculated using JMP statistical analysis software (SAS, NC, USA). Differences in the data were considered significant at $P < 0.05$.

## Results

### Initial screening of the chemical compound library

A total of 23,227 chemical compounds were evaluated by cultivation in 96-well plates as an initial screening process (Fig 1). In the rCHO cell cultures, 566 compounds showed a positive effect in terms of increased mAb production. The hit rate was 2.5%. In parallel, we investigated the commercial availability of the first screening positives before starting the second screening to obtain sufficient amounts of the positives for further study. Although 67 chemical compounds were commercially available, another 5 compounds for which the original structures were not commercially available were also tested as derivatives. Therefore, a total of 72 samples underwent the second screening culture.

### Second screening of the selected compounds

Seventy-two chemical compounds were further evaluated. The relative cell-specific productivity, relative mAb concentration, and viability were assessed to evaluate the chemical compound treatments against the control condition. The selection criteria of the candidate compounds were set to over 120% for relative mAb concentration, 105% for relative cell-specific productivity, and 80% for viability. The samples with the ID numbers 42, 62, and 67 met

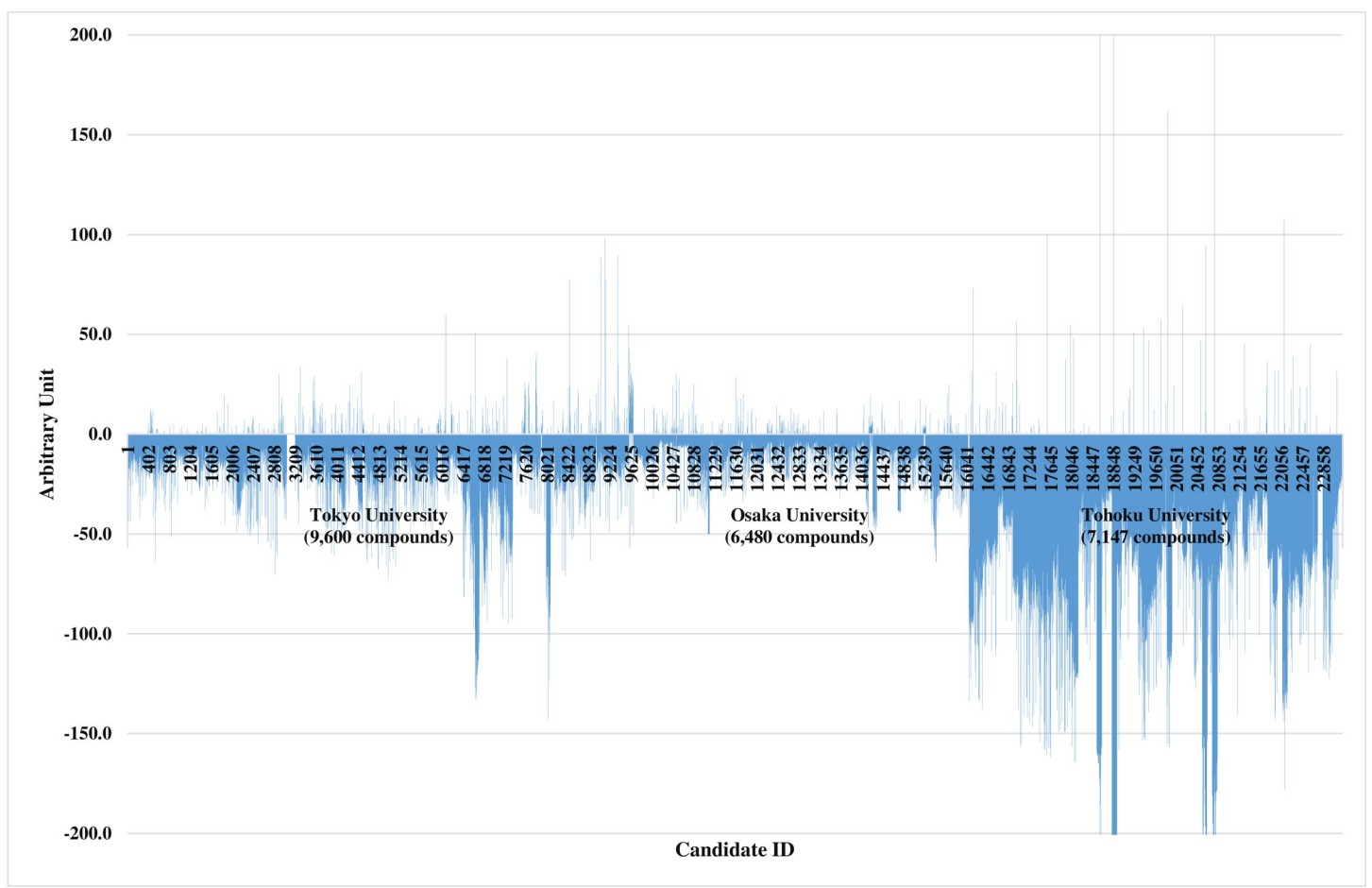

**Fig 1. Results of the first screening to evaluate the effect of the chemical compounds on mAb production.** Higher mAb production is shown as a positive arbitrary unit.

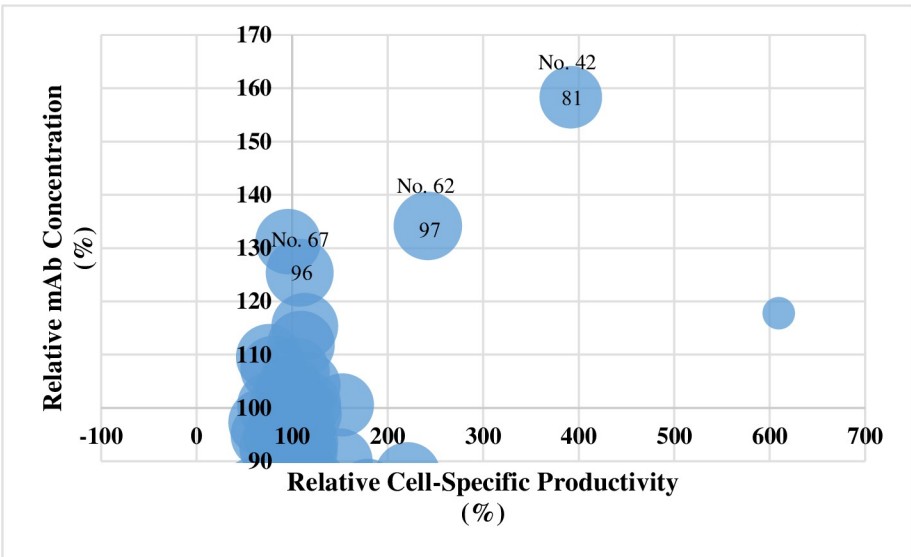

**Fig 2. Relative mAb concentration and cell-specific productivity increased with the addition of the chemical compounds.** The numbers in and above the circles indicate viability and the sample's ID number, respectively.

these criteria, with increases of 125 to 158% for mAb concentration and 108 to 392% for cell-specific productivity in comparison with the control condition (Fig 2).

**MPPB as an effective additive of mAb production.** The samples with the ID numbers 42, 62, and 67 were re-evaluated to confirm the reproducibility of second screening results. The three selected compounds had higher relative mAb concentration and relative cell-specific productivity. ID number 62 showed an especially strong effect, with a 171% increase in relative mAb concentration and a 202% increase in relative cell-specific productivity compared with the control condition (Fig 3). As ID number 62 demonstrated the largest effect, we further investigated the characteristics of ID number 62 in batch and fed-batch cultures. The selected compound was MPPB (Fig 4), a commercially available substitute for a first-screening-hit-compound supplied from the chemical compound library of the University of Tokyo. MPPB was originally developed as an anti-tuberculosis therapeutic compound [25–27].

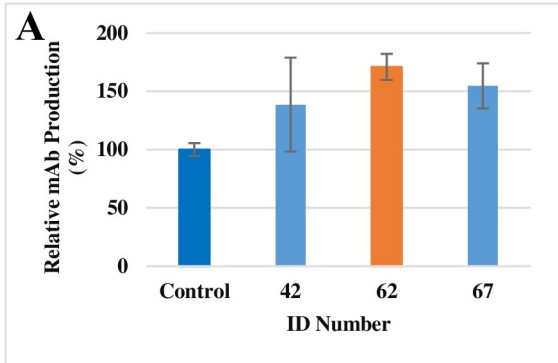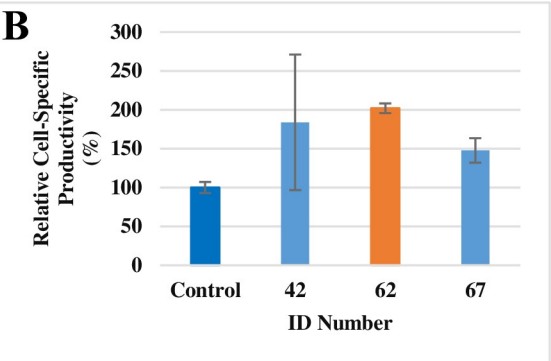

**Fig 3. The effect of the three candidate compounds on relative mAb concentration and relative cell-specific productivity.** All cell cultures were executed three times and evaluated on day 3. Each value of relative mAb production (**A**) and relative cell-specific productivity (**B**) were compared statistically.

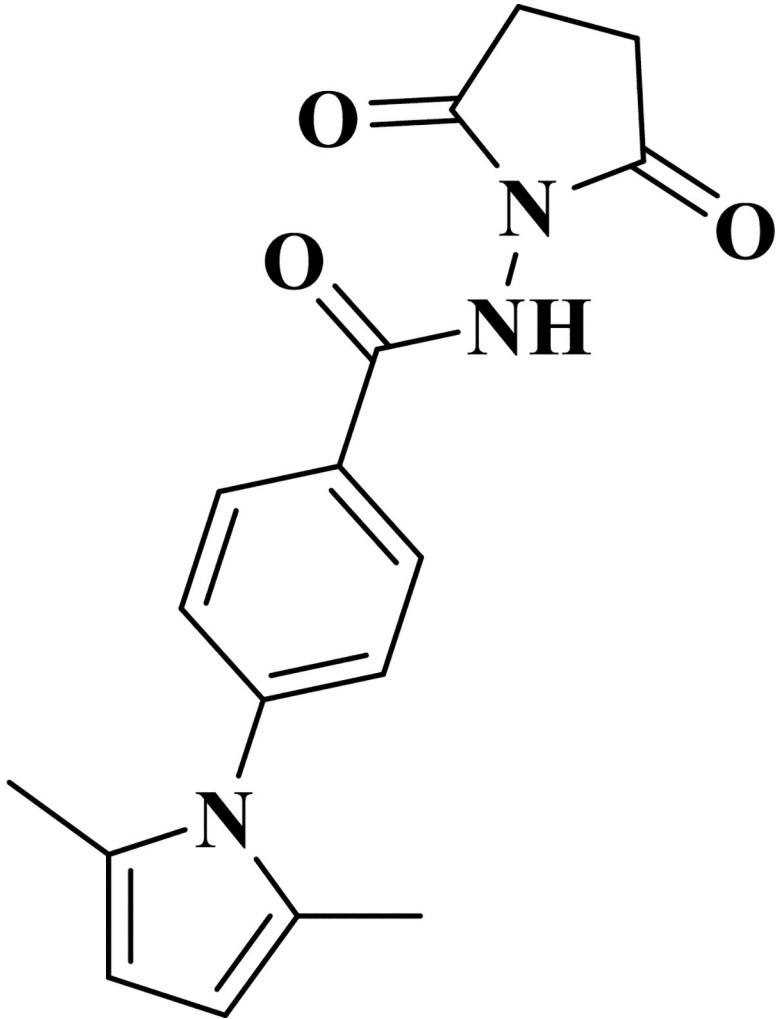

**Fig 4. The structure of 4-(2,5-dimethyl-1*H*-pyrrol-1-yl)-*N*-(2,5-dioxopyrrolidin-1-yl) benzamide (MPPB).**

### Effect of MPPB on the culture profile in the batch cultures

The effect of MPPB on the culture profile was tested in the batch cultures, as shown in Fig 5. MPPB suppressed cell growth depending on the concentration of MPPB added (Fig 5A). Viability was slightly decreased at a MPPB concentration of 1.28 mM (Fig 5B) and cell-specific productivity was increased by an addition of MPPB with a concentration over 0.32 mM (Fig 5C). A dose-dependent reduction in day 4 mAb concentration was observed with increasing concentrations of MPPB due to suppressed cell growth in these short-period batch cultures (Fig 5D). These results showed that treatment with MPPB concentrations of 0.32 to 0.64 mM suppressed cell growth and increased cell-specific productivity while retaining viability. Therefore, MPPB concentrations of 0.32 to 0.64 mM were used to characterize MPPB.

Becker et al. [30] and Hara and Kondo [31] reported that increased amounts of intracellular ATP are associated with increased cell-specific productivity. Therefore, we investigated intracellular ATP amounts and cell-specific glucose uptake rates, which are related to ATP production [32, 33], in the batch cultures to clarify the reason for the increasing cell-specific productivity (Fig 6). The results showed that both intracellular ATP amounts and cell-specific glucose uptake rates were increased with higher MPPB concentrations. Intracellular ATP

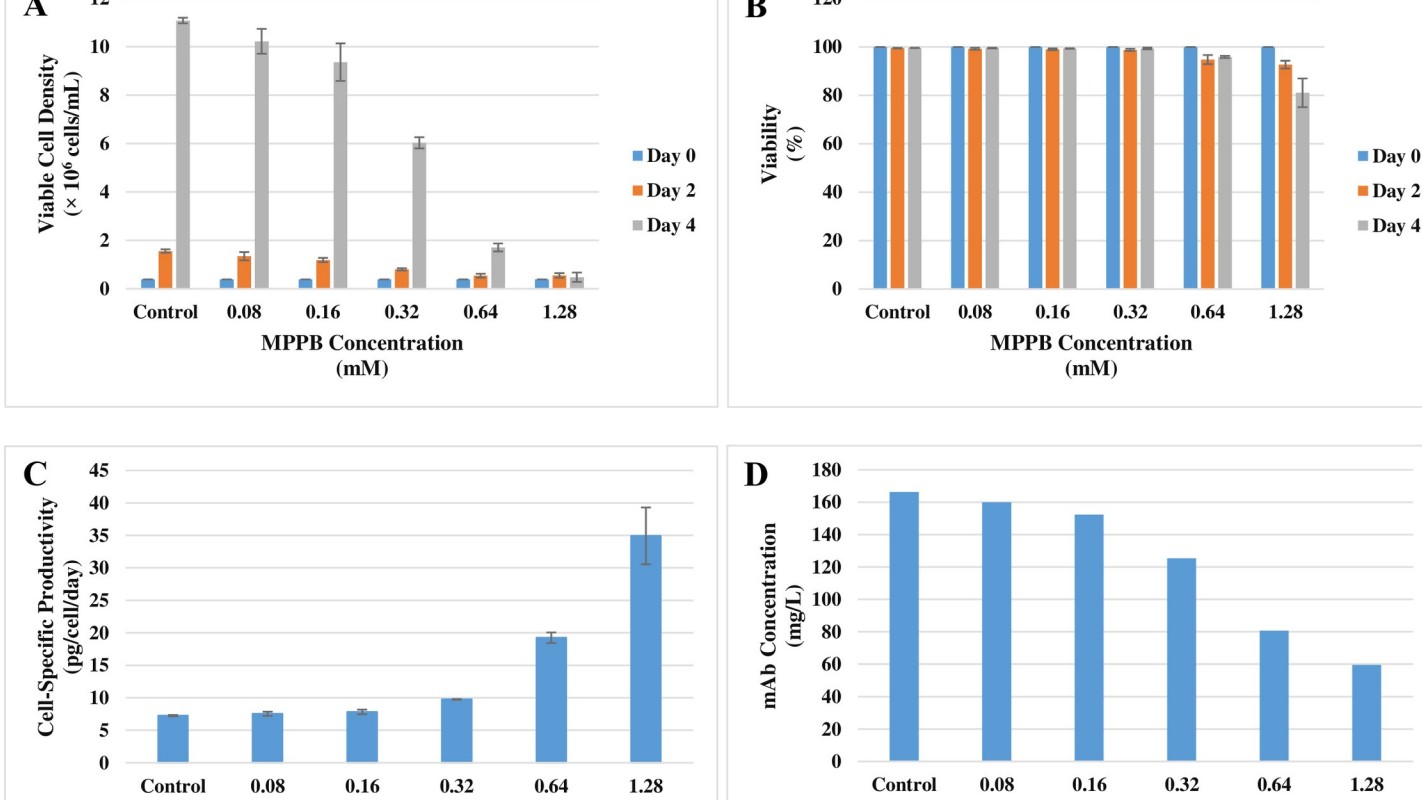

**Fig 5. MPPB suppresses cell growth and increases cell-specific productivity.** The tested MPPB concentrations are indicated in the figure. In these tests, viable cell density (**A**), viability (**B**), cell-specific productivity on day 4 (**C**), mAb concentration on day 4 (**D**) were evaluated.

amounts increased from 8 to 24 fmol/cell (Fig 6A), and cell-specific glucose uptake rates increased from 0.9 to 2.6 pmol/cell/day (Fig 6B). Intracellular ATP amounts and cell-specific glucose uptake rates were positively correlated with cell-specific productivity (Fig 6C). Additionally, cell growth was suppressed with MPPB treatment (Fig 6D)

MPPB was further evaluated in the batch cultures. These cell cultures were continued until less than 70% viability was reached. Culture conditions including pH, glucose concentration, and aeration were not optimized in any cell culture at this time in order to evaluate the effect of MPPB only. The maximum VCD was reduced from $16.6 \times 10^6$ cells/mL to $8.0 \times 10^6$ cells/mL with MPPB treatment (Fig 7A). On the other hand, the MPPB-added condition retained higher viability (Fig 7B). The mAb concentration under the MPPB-added condition was almost the same value as the control condition at each harvest time point (Fig 7C). The MPPB addition increased the cell-specific productivity from 4.2 pg/cell/day to 7.9 pg/cell/day (Fig 7D).

From a spent media analysis, glucose and lactate concentrations showed no differences between the conditions with and without the addition of MPPB (Fig 8A and 8B). However, cell-specific glucose uptake rate increased from 0.49 to 1.1 pmol/cell/day with MPPB treatment (Fig 8C). Meanwhile, the cell-specific lactate production rate increased from 0.22 to 0.37 pmol/cell/day under the MPPB-added condition (Fig 8D). The increase in the cell-specific

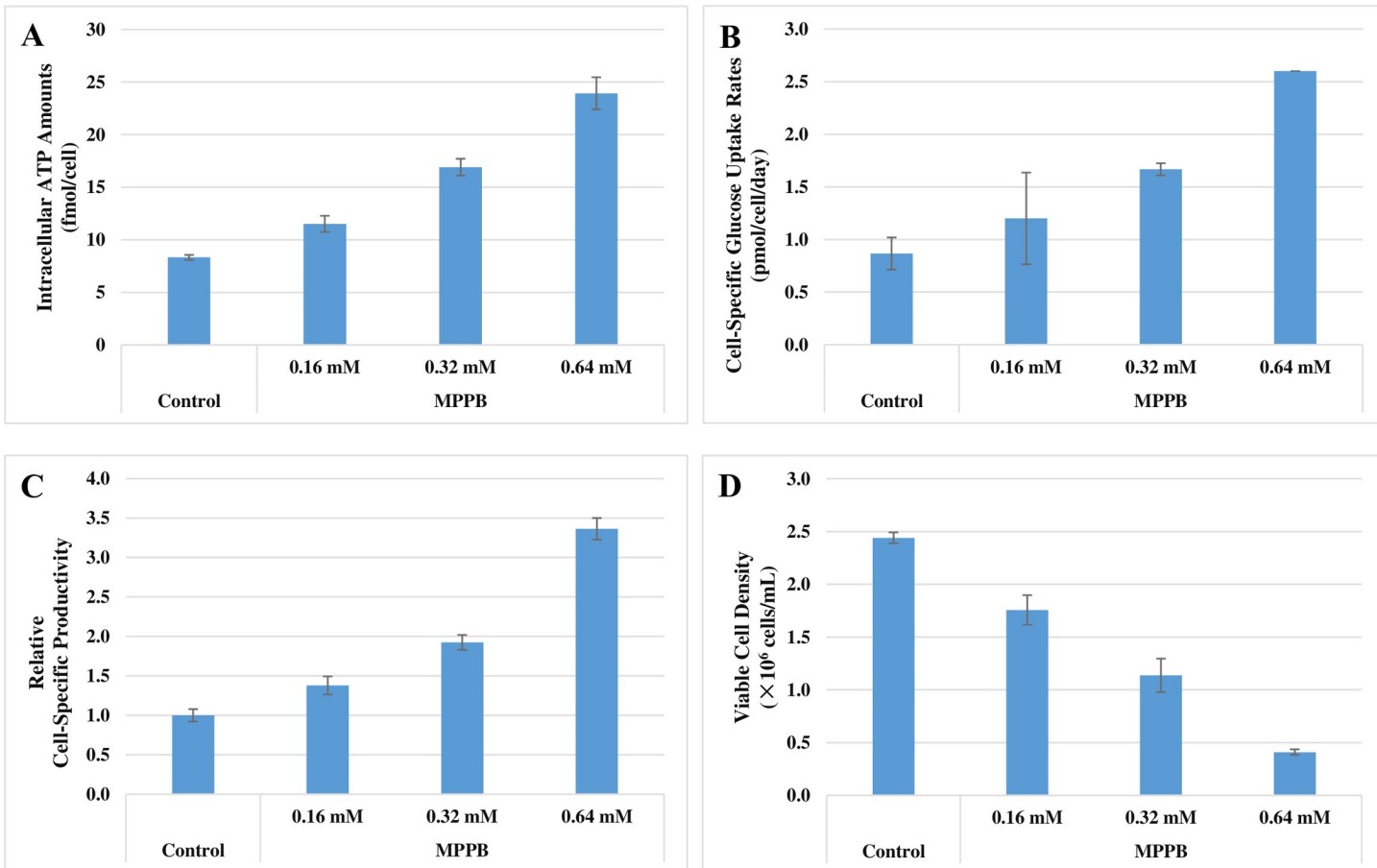

**Fig 6. MPPB increased both intracellular ATP amounts and cell-specific glucose uptake rates and suppressed cell growth.** Given previous results, the tested MPPB concentrations ranged from 0.32 to 0.64 mM. Intracellular ATP amounts (**A**), cell-specific glucose uptake rates (**B**), relative cell-specific productivity (**C**), and viable cell density (**D**) were evaluated on day 3.

lactate production rate was 0.15 pmol/cell/day, which was smaller than the increase in the cell-specific glucose uptake rate (0.61 pmol/cell/day). This result suggests that consumed glucose was more efficiently converted to ATP via the tricarboxylic acid (TCA) cycle in the MPPB-added condition.

## Effect of MPPB on the culture profile in the fed-batch cultures

Further evaluation of the MPPB-added condition was continued in the fed-batch cultures. These cell cultures were continued until less than 70% viability was reached. Culture conditions (pH, glucose concentration, and aeration) were not optimized in any cell culture at this time in order to evaluate the effect of MPPB only. The maximum VCD under the MPPB-added condition was reduced from $21.2 \times 10^6$ cells/mL in the control condition to $14.0 \times 10^6$ cells/mL (Fig 9A). The VCD and viability also decreased after day 8 in the control condition (Fig 9A and 9B). However, they were maintained until day 12 under the MPPB-added condition. Despite the lower VCD, the final mAb concentration under the MPPB-added condition reached 1,098 mg/L which was 1.5-folds higher than that under the control condition (Fig 9C). The cell-specific productivity under the MPPB-added condition was also 1.5-fold higher than that under the control condition (Fig 9D). The cell-specific productivity under the control and MPPB-added conditions was 7.1 and 11 pg/cell/day, respectively.

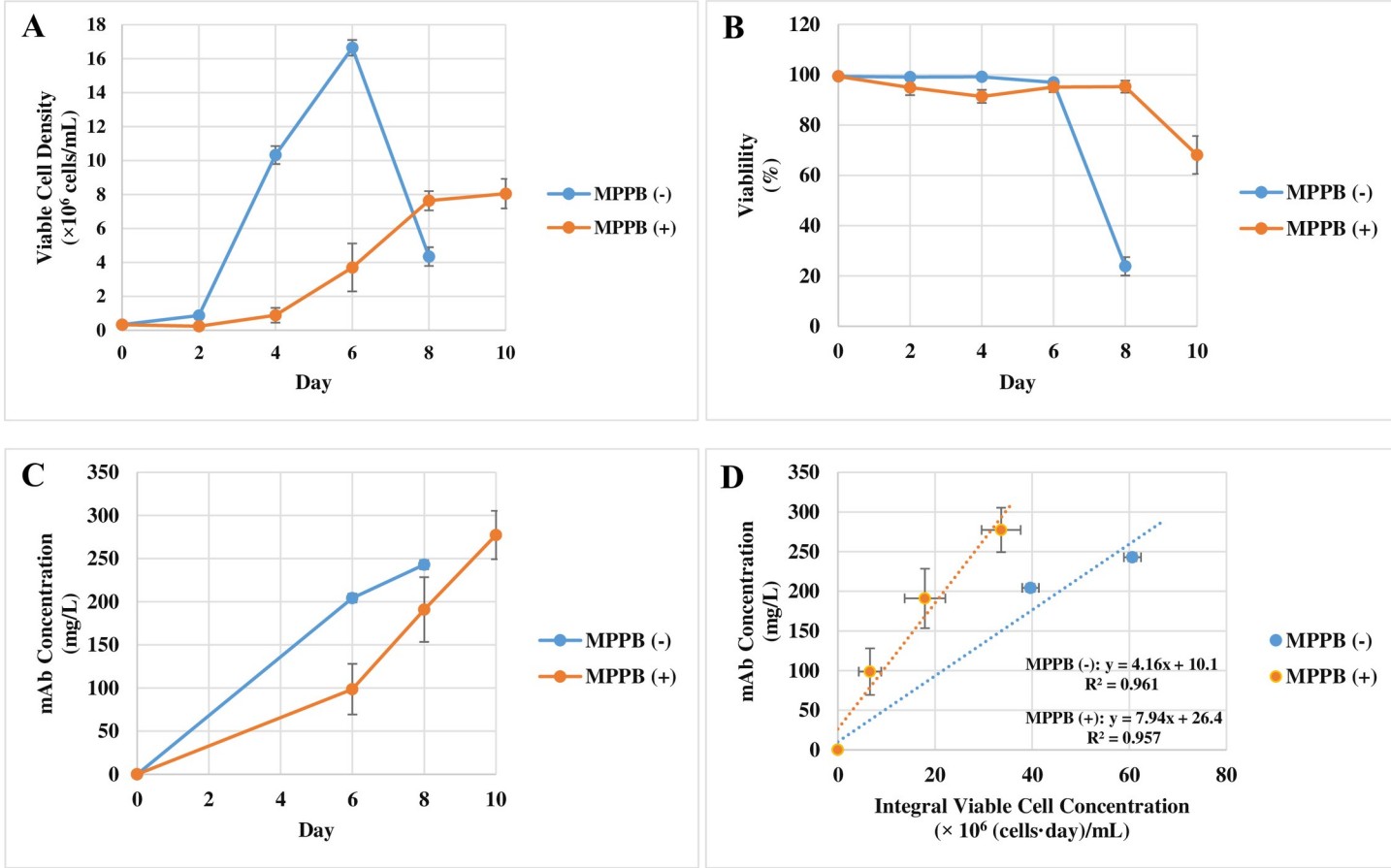

**Fig 7. Cell culture profile and mAb production in rCHO cells treated with MPPB.** MPPB at 0.64 mM was added in the batch cultures on day 0. Viable cell density (**A**) and viability (**B**) were measured every 2 days. Cell culture was continued until viability below 70% was reached. The slope of mAb concentration (**C**) to integral viable cell concentration was used to indicate cell-specific productivity (**D**) [28].

Residual glucose was fully consumed by day 10 under the control condition, while glucose under the MPPB-added condition was retained until day 14 (Fig 10A). The cell-specific glucose uptake rates were 0.63 and 0.74 pmol/cell/day under the control and MPPB-added conditions, respectively (Fig 10C). The lactate concentration was kept under 0.8 g/L, and the cell-specific lactate production rates were similar in both conditions (Fig 10B and 10D).

In addition, the major N-linked glycans (G0F, G1F, G2F, and M5) of mAb were investigated on each final culture day (MPPB (+): day 14, MPPB (-): day 10). G0F was the major N-linked glycan, and G1F was decreased from 24.5 to 14.8% under the MPPB-added condition, although G2F and M5 ratios were not changed, as shown in Fig 11.

### Structure-activity relationship of MPPB

The MPPB structure is divided into five chemical components, as follows: N-(2,5-dioxopyrrolidin-1-yl) benzamide; 4-(2,5-dimethyl-1H-pyrrol-1-yl) benzamide; succinimide; 4-aminobenzamide; and 2,5-dimethylpyrrole (Fig 12).

The activity of the five chemical components derived from MPPB was identified and compared with that of MPPB in terms of cell-specific productivity in the batch cultures (Fig 13). 4-(2,5-Dimethyl-1H-pyrrol-1-yl) benzamide and 2,5-dimethylpyrrole increased cell-specific productivity. Further, the 2,5-dimethylpyrrole-added condition had a 2.2-fold higher cell-specific

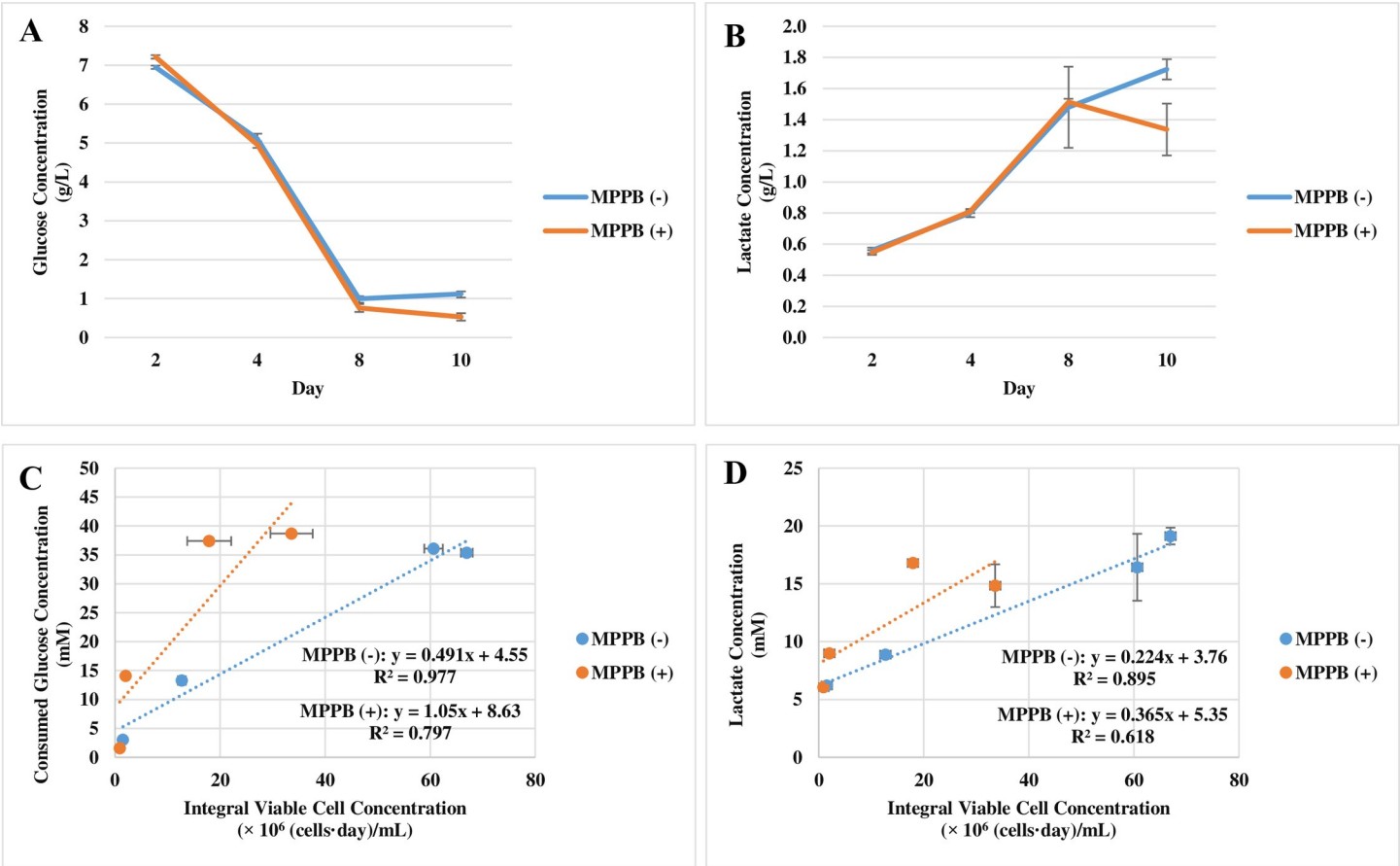

**Fig 8. The effect of MPPB on the metabolite profile in the batch cultures.** The spent media analysis of glucose concentration (**A**) and lactate concentration (**B**) was conducted every 2 days starting on day 2. Cell-specific glucose uptake rates (**C**) and cell-specific lactate production rates (**D**) were indicated as the slopes of consumed glucose concentration and lactate concentration, respectively to integral viable cell concentration [28, 29].

productivity than the control condition. These results suggested that 2,5-dimethylpyrrole was the most active chemical structure of MPPB.

Further evaluation was conducted to identify the effect of pyrrole derivatives on viability and cell-specific productivity in the batch cultures (Fig 14, Table 1). Cell-specific productivity with alkyl pyrrole derivatives (**2**, **9–14**) was increased up to 1.4- to 7.8-fold higher than that with the control condition, although pyrrole (**7**) and 1-alkyl pyrroles (**1**, **8**) did not show any activity. On the other hand, viability decreased below 50% with these alkyl pyrrole derivatives (**2**, **9–14**), except for 2,5-dimethylpyrrole (**13**). 2,5-Dimethylpyrrolidine (**6**), which has a reduced framework of 2,5-dimethylpyrrole (**13**), also did not affect the cell culture. In addition, the alkyl pyrroles (**9–14**) that showed high cell-specific productivity in the 0.32 mM concentration were also tested in other MPPB concentrations (S2 Fig). The results showed that 2,5-dimethylpyrrole (**13**) effectively increased cell-specific productivity without decreasing viability. Thus, 2,5-dimethylpyrrole (**13**) was found to be the most effective chemical structure to increase cell-specific productivity while retaining viability.

## Discussion

In this study, we screened chemical compounds to find novel additives that improve mAb production in rCHO cells and thereby promote pharmaceutical supply (Figs 1–3). MPPB had

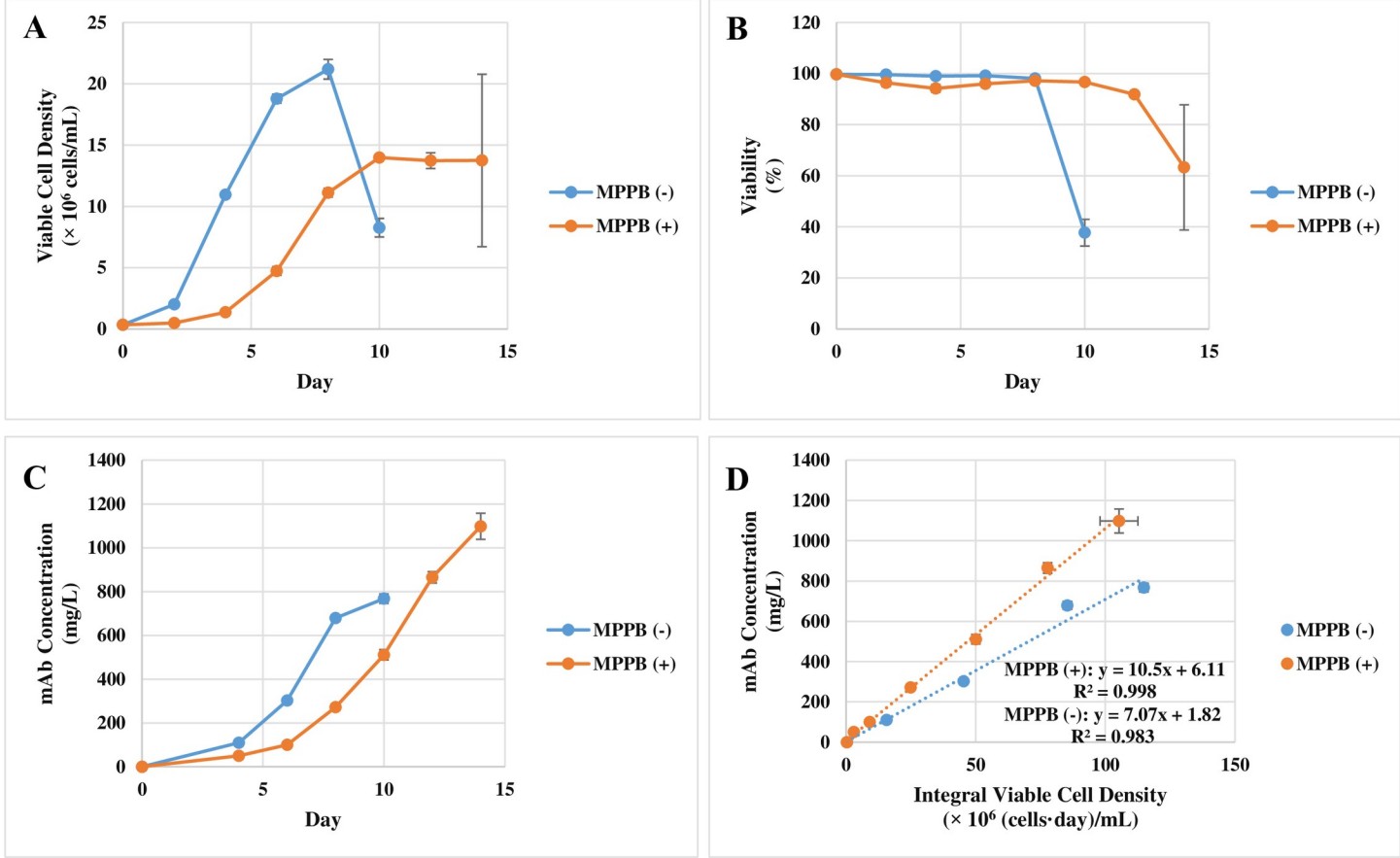

**Fig 9. Cell culture profile and mAb production in the fed-batch cultures under the MPPB-added condition.** MPPB at 0.64 mM was added to a culture on day 0, and 2% (v/v) feed medium was added on days 2, 4, and 6. Viable cell density (**A**) and viability (**B**) were measured every 2 days. The cell culture was continued until viability below 70% was reached. The slope of the mAb concentration (**C**) and integral viable cell concentration were used as cell-specific productivity (**D**) [28].

1.5-fold higher mAb production than the control conditions in the fed-batch cultures. This is the first study in which MPPB was applied in rCHO cell cultures to improve mAb production.

Our study revealed two key factors for the improvement of mAb production by MPPB: retaining high viability (Figs 7B and 9B), and increasing of cell-specific productivity (Figs 7D and 9D). It is likely that these two factors explain the increased mAb production in the MPPB-added condition despite lower cell concentrations compared to the control condition.

The mechanism of MPPB in terms of retaining viability is unclear. MPPB might keep glucose concentrations high, leading to retained viability (Figs 9B and 10A). However, although the remaining glucose concentration was almost similar with and without the addition of MPPB in the batch cultures, the viability remained high under the MPPB-added condition (Figs 7B and 8A). Once, glucose was maintained above 1 g/L in the fed-batch cultures (S1 Fig), the viability under the control condition was 45% lower than that under the MPPB-added condition on day 12. These results suggested that cell growth suppression by MPPB treatment prevented the depletion of medium components except for glucose, resulting in maintained viability.

The cause of increased cell-specific productivity, however, is clearer. In our study, MPPB treatment had two important effects to increase cell-specific productivity. One was the

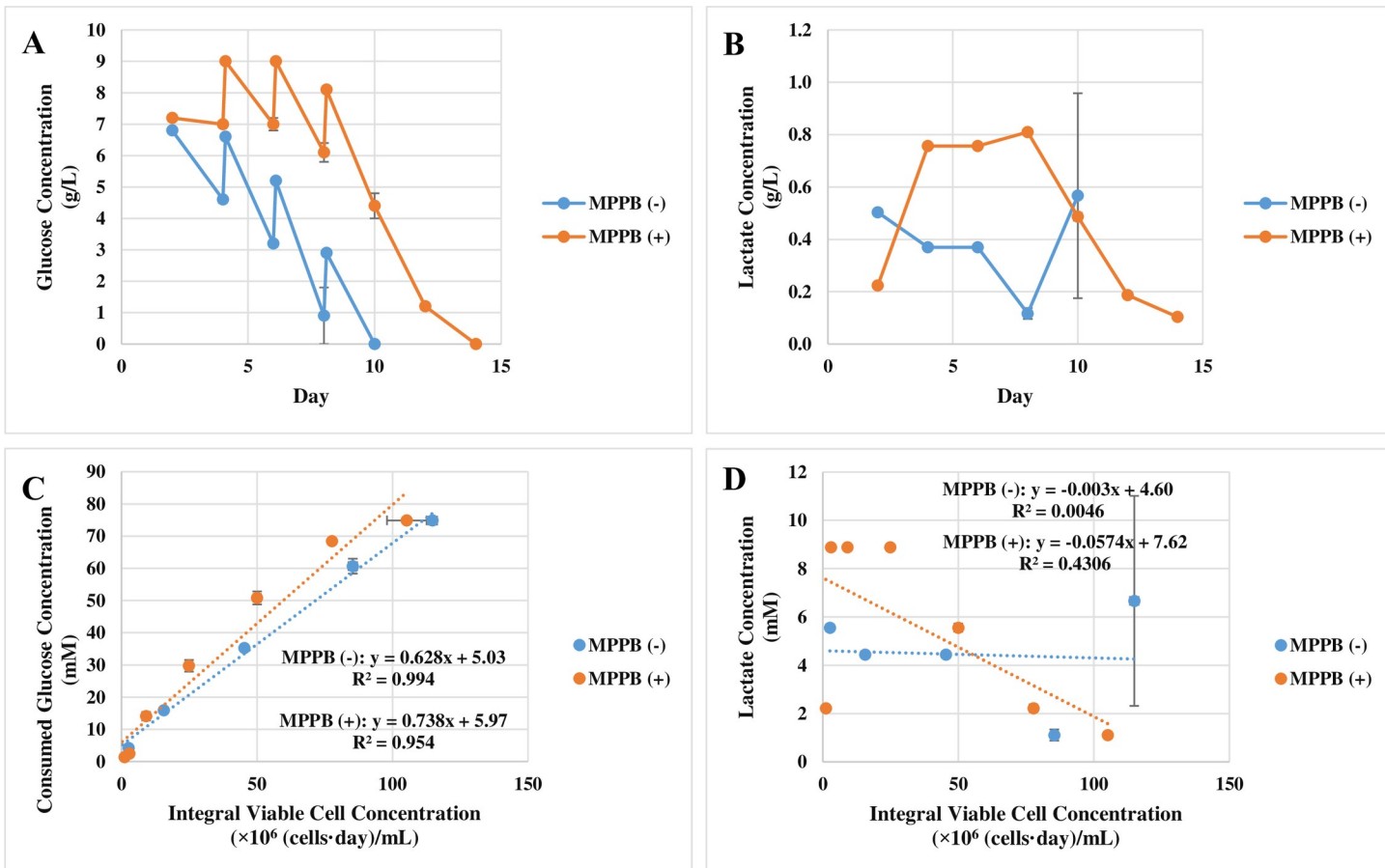

**Fig 10. Metabolite analysis of the fed-batch cultures under the MPPB-added condition.** Glucose concentration (**A**) and lactate concentration (**B**) were measured before adding the feed medium. Glucose concentration after adding the feed medium was calculated using the added feed medium amount. Cell-specific glucose uptake rates (**C**) and cell-specific lactate production rates (**D**) were indicated as the slopes of consumed glucose concentration and lactate concentration, respectively, against integral viable cell concentration [28, 29].

suppression of cell growth (Figs 5A, 6D, 7A and 9A), and the other was the acceleration of cell-specific glucose uptake rates (Figs 6B, 8C and 10C), while keeping cell-specific lactate production rates low (Figs 8D and 10D). Both the effects would control the amounts of intracellular ATP. As reported by Mulukutla et al. [34], growing cells require more ATP than quiescent cells. Templeton et al. [35] reported that the stationary phase, which is a suppressed state of cell growth, activates the TCA cycle. An increased cell-specific glucose uptake rate leads to an increased amounts of intracellular ATP when glucose is metabolized to ATP in the TCA cycle [32, 33], but not when glucose is metabolized to lactate. Becker et al. [30] and Hara and Kondo [31] reported that the energy available in the form of intracellular ATP is crucial for mAb production. Accordingly, it may be hypothesized that rCHO cells increased intracellular ATP amounts by suppressing cell growth and increasing cell-specific glucose uptake rates in MPPB-supplemented condition, thereby increasing cell-specific productivity, as expected. To test this hypothesis, we investigated the relationship between the amount of intracellular ATP and MPPB concentration (Fig 6). We found that the amount of intracellular ATP was enhanced with higher MPPB concentrations (Fig 6A). Cell-specific glucose uptake rates and cell-specific productivity were also increased (Fig 6B and 6C). On the other hand, cell growth was also

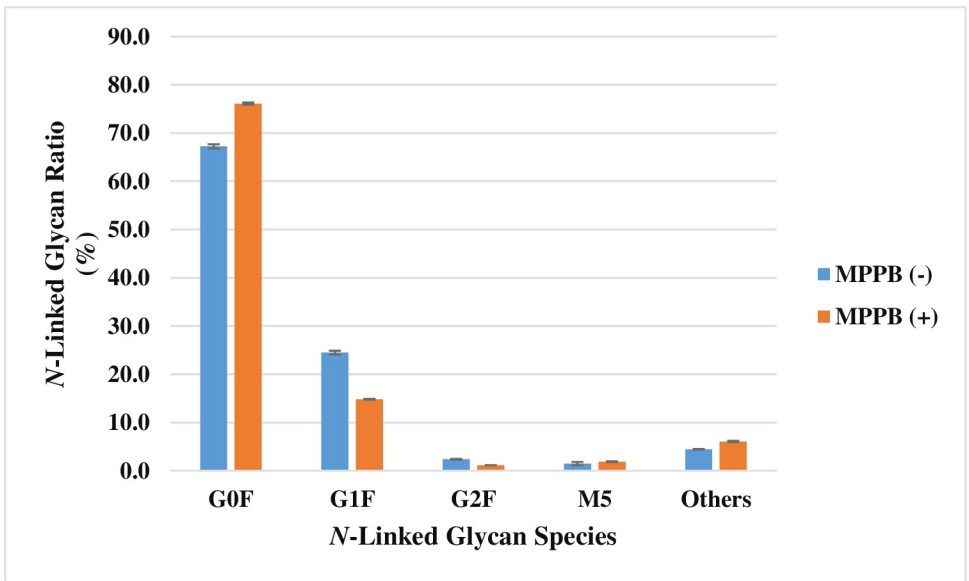

**Fig 11. MPPB suppresses galactosylation of mAb.** Each value was calculated as the percentage of total *N*-linked glycans. The ratio was statically analyzed.

suppressed with higher MPPB concentrations (Fig 6D). Therefore, we predicted that the increased cell-specific productivity under the MPPB-added condition might occur due to the increasing amount of intracellular ATP resulting from cell growth suppression and enhanced cell-specific glucose uptake rates. Our results showed that a chemical compound such as MPPB, which suppresses cell growth and increases intracellular ATP amounts, may improve mAb production in rCHO cell cultures.

Furthermore, we found that MPPB treatment suppressed galactosylation in *N*-linked glycans on the expressed mAb (Fig 11). The galactosylation ratio in *N*-linked glycans on mAb is

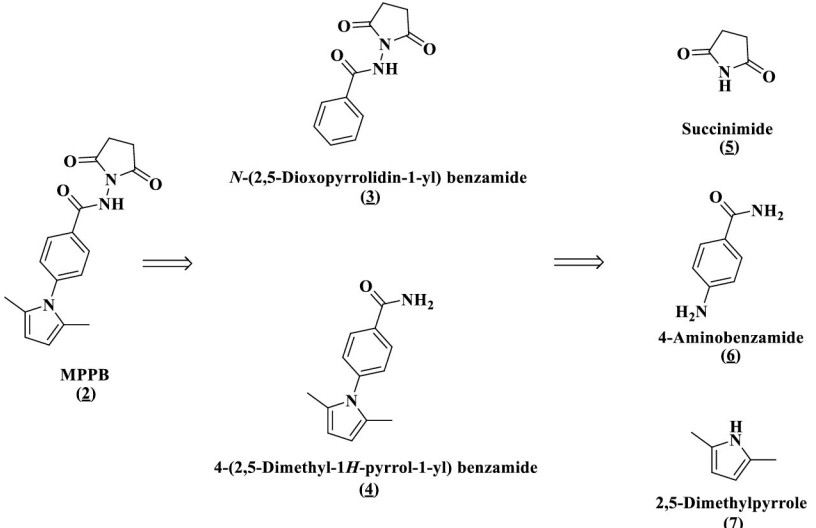

**Fig 12. Five chemical components derived from MPPB.**

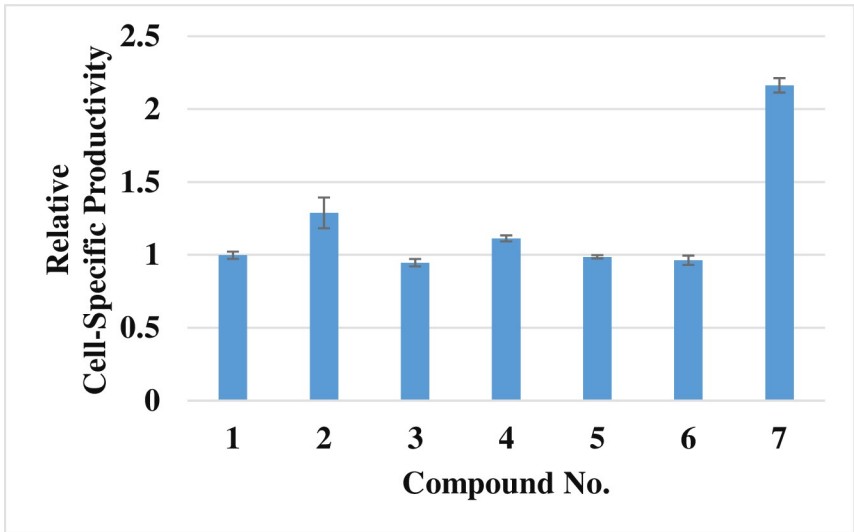

**Fig 13. Evaluation of the effect of the five chemical components derived from MPPB on cell-specific productivity.**
Each chemical compound was added on day 0 and evaluated on day 3. The tested concentration of each chemical compound was 0.32 mM. Lane 1: dimethyl sulfoxide (control); lane 2: MPPB; lane 3: *N*-(2,5-dioxopyrrolidin-1-yl) benzamide; lane 4: 4-(2,5-dimethyl-1*H*-pyrrol-1-yl) benzamide; lane 5: succinimide; lane 6: 4-aminobenzamide; lane 7: 2,5-dimethylpyrrole.

one of the critical quality attributes of mAb in terms of pharmaceutical development, because the galactosylation profile affects complement-dependent cytotoxicity [36, 37] and antibody-dependent cellular cytotoxicity [38–40]. Therefore, to adjust mAb activity and maintain consistent quality, galactosylation is often controlled during pharmaceutical process research and development. Some media additives are known to affect the galactosylation of mAb in rCHO cells. For example, valeric acid, valproic acid, CDK4/6 inhibitor, galactose, manganese, and uridine encourage galactosylation [18, 19, 22, 41, 42], while zinc and ammonia suppress galactosylation [43, 44]. However, ammonia is harmful to rCHO cells [45]. The number of effective galactsylation suppressants is small, and, therefore, the suppression of galactosylation by media additives is not well understood. Our findings that MPPB suppressed the galactosylation may offer a new option to control the quality of mAb by the suppression of galactosylation in the *N*-linked glycans of mAb.

Finally, the structure-activity relationship study revealed that the most critical chemical portion of MPPB to maintain the viability and improve the cell-specific productivity of mAb in rCHO cells was 2,5-dimethylpyrrole (Figs 13 and 14, Table 1). Since 2,5-dimethylpyrrolidine having a reduced framework of 2,5-dimethylpyrrole showed no effect, the importance of the heteroaromatic framework of 2,5-dimethylpyrrole for both the activities was suggested. The 2,5-dialkyl substitution of pyrrole was also indispensable. The viability was decreased in the cases of other mono- or bis-alkyl-substituted pyrroles. Furthermore, non-alkylated or *N*-substituted pyrroles showed no effect on both the viability and the cell-specific productivity. Thus, the 2,5-dialkyl-substituted pyrrole framework was found to be effective on maintaining viability and improving cell-specific productivity of mAb in rCHO cells during the structure-activity relationship study. The mechanism of the action of MPPB and 2,5-dimethylpyrrole in rCHO cell cultures is currently being considered for future application studies.

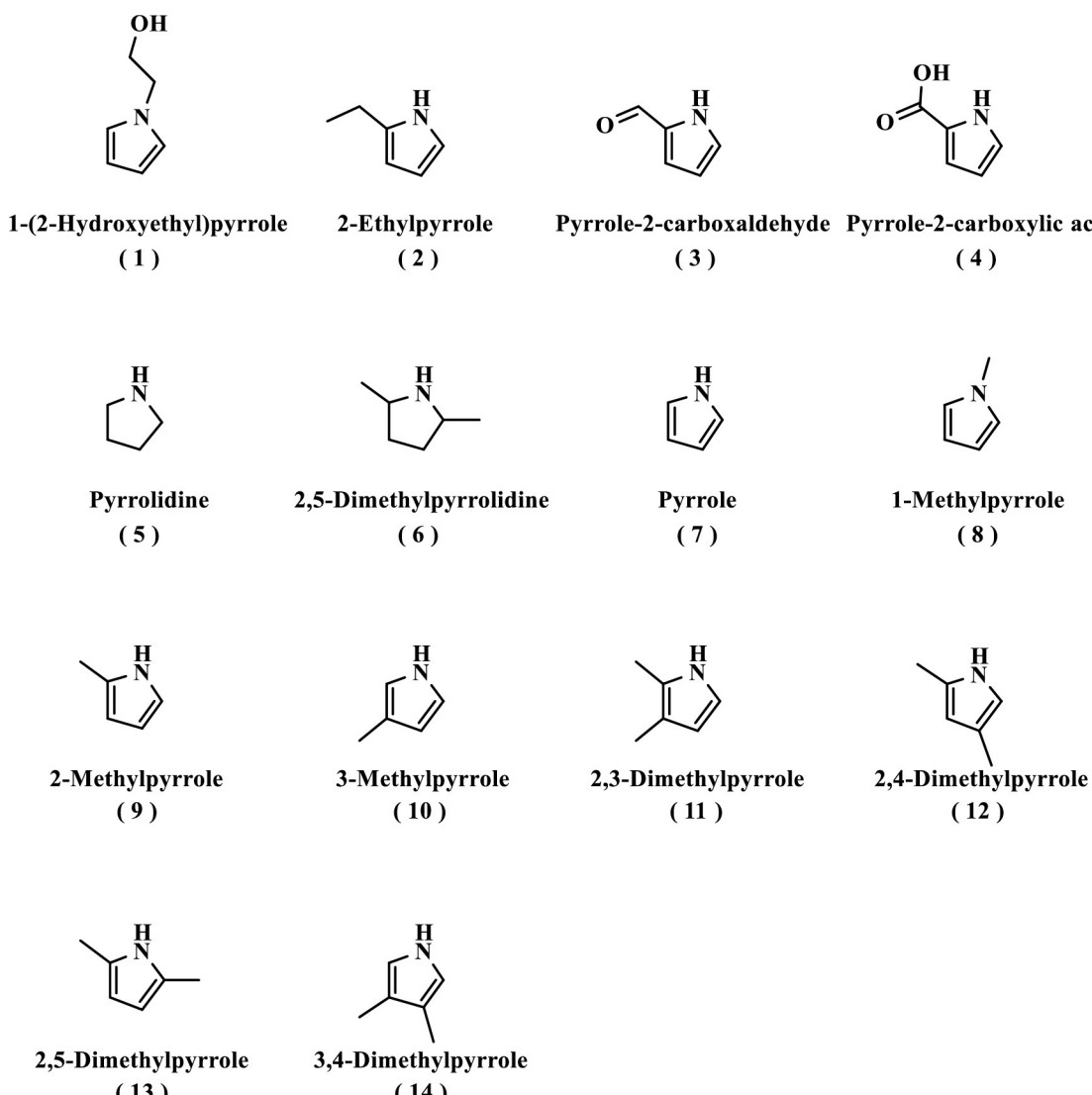

**Fig 14. Additional pyrrole derivatives tested to identify the structure-activity relationship of MPPB with viability and relative cell-specific productivity in rCHO cell cultures.**

**Table 1. Effect of pyrrole derivatives on viability and relative cell-specific productivity in rCHO cell cultures.**

| Compound No. | Compound name | Viability (%) | Relative cell-specific productivity |
|---|---|---|---|
| - | Dimethyl sulfoxide (control) | 99.4 (0.4) | 1.0 (0.0) |
| - | MPPB | 99.2 (0.5) | 1.4 (0.2) |
| 1 | 1-(2-Hydroxyethyl)pyrrole | 99.6 (0.2) | 1.0 (0.1) |
| 2 | 2-Ethylpyrrole | 41.2 (2.5) | 1.4 (0.1) |
| 3 | Pyrrole-2-carboxaldehyde | 99.3 (0.2) | 1.0 (0.1) |
| 4 | Pyrrole-2-carboxylic acid | 99.5 (0.2) | 1.0 (0.1) |
| 5 | Pyrrolidine | 98.9 (1.4) | 1.0 (0.0) |
| 6 | 2,5-Dimethylpyrrolidine | 99.0 (1.0) | 1.0 (0.0) |
| 7 | Pyrrole | 99.3 (0.7) | 1.0 (0.1) |

*(Continued)*

**Table 1.** (Continued)

| Compound No. | Compound name | Viability (%) | Relative cell-specific productivity |
|:---:|:---:|:---:|:---:|
| 8 | 1-Methylpyrrole | 99.4 (0.1) | 1.0 (0.1) |
| 9 | 2-Methylpyrrole | 50.2 (0.2) | 3.0 (0.2) |
| 10 | 3-Methylpyrrole | 34.4 (2.0) | 4.0 (0.2) |
| 11 | 2,3-Dimethylpyrrole | 2.6 (1.7) | 7.8 (0.4) |
| 12 | 2,4-Dimethylpyrrole | 30.7 (3.4) | 4.8 (0.3) |
| 13 | 2,5-Dimethylpyrrole | 97.9 (0.6) | 2.2 (0.0) |
| 14 | 3,4-Dimethylpyrrole | 4.0 (1.2) | 7.4 (0.3) |

Each pyrrole derivative was added on day 0 and evaluated on day 3. The tested concentration of each pyrrole derivative was 0.32 mM. Standard deviation was indicated in the blanket.

## Supporting information

**S1 Fig. The fed-batch culture profile in the retained glucose condition.** Glucose was added on days 8 and 12 to keep the concentration above 1 g/L in the fed-batch cultures. And Feed medium was added on days 4, 6, and 8. Viable cell density (**A**), viability (**B**), glucose concentration (**C**), lactate concentration (**D**), and mAb concentration (**E**) were measured every 2 days. The each slope of mAb concentration, consumed glucose concentration and lactate concentration to integral viable cell concentration were used to indicate cell-specific productivity (**F**), cell-specific glucose uptake rates (**G**), and cell-specific lactate production rates (**H**), respectively [28].
(DOCX)

**S2 Fig. The trend of viability and cell-specific productivity in each pyrrole derivatives concentration.** Viability (**A**) and cell-specific productivity (**B**) were analyzed by sampling day 3 cells in the batch cultures. The tested pyrrole derivative concentrations were 0.08 to 0.64 mM in the culture solutions. Culture conditions were considered adequate when viability was above 80%.
(DOCX)

**S1 Table. The validity of the first screening condition.** DMSO was used as the 0% control, and 4-phenyl butyric acid dissolved in DMSO was used as the 100% control. Validity was evaluated at the points of coefficient of variation (CV) using day 3 cell culture solution.
(DOCX)

## Acknowledgments

The authors acknowledge the support provided by Akihiro Kawara (Daiichi Sankyo Co., Ltd.) in terms of contracting with the BINDS program, and by Masami Katase (Daiichi Sankyo Co., Ltd.) in terms of collecting commercially available chemical compounds.

## Author Contributions

**Conceptualization:** Yuichi Aki.

**Data curation:** Yuichi Aki.

**Formal analysis:** Yuichi Aki.

**Investigation:** Yuichi Aki, Yuta Katsumata.

**Methodology:** Yuichi Aki.

**Project administration:** Yuichi Aki.

**Resources:** Yuichi Aki.

**Supervision:** Hirofumi Kakihara, Koichi Nonaka, Kenshu Fujiwara.

**Validation:** Yuichi Aki.

**Visualization:** Yuichi Aki, Koichi Nonaka.

**Writing – original draft:** Yuichi Aki, Koichi Nonaka.

**Writing – review & editing:** Yuichi Aki, Yuta Katsumata, Hirofumi Kakihara, Koichi Nonaka, Kenshu Fujiwara.

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
