## [Decision Letter · Decision Letter 0]

8 Dec 2020

PONE-D-20-26236

4-(2,5-Dimethyl-1H-pyrrol-1-yl)-N-(2,5-dioxopyrrolidin-1-yl) benzamide improves monoclonal antibody production in a Chinese hamster ovary cell culture

PLOS ONE

Dear Dr. Aki,

Thank you for submitting your manuscript to PLOS ONE. After careful consideration, we feel that it has merit but does not fully meet PLOS ONE’s publication criteria as it currently stands. Therefore, we invite you to submit a revised version of the manuscript that addresses the points raised during the review process.

Please carefully consider all comments by both reviewers including those detailed comments of grammer mistakes marked in the attached manuscript file. 

We look forward to receiving your revised manuscript.

Kind regards,

Yong-Bin Yan, Ph.D.

Academic Editor

PLOS ONE

Journal Requirements:

2.We note that you have indicated that data from this study are available upon request. PLOS only allows data to be available upon request if there are legal or ethical restrictions on sharing data publicly. For more information on unacceptable data access restrictions, please see http://journals.plos.org/plosone/s/data-availability#loc-unacceptable-data-access-restrictions.

3.Thank you for stating the following in the Acknowledgments Section of your manuscript:

[This research was supported by the Platform Project for Supporting Drug Discovery and Life

Science Research (Basis for Supporting Innovative Drug Discovery and Life Science Research [BINDS])

from Japan Agency for Medical Research and Development (AMED) under grant numbers

JP19am0101086, JP19am0101084, and JP19am0101095 (support numbers 1009, 0847, and 1011).]

 [The authors received no specific funding for this work.

As a notice, in this study, chemical compounds library was collected from  Basis for Supporting Innovative Drug Discovery and Life Science Research (https://www.binds.jp/) for free. But, it didn't affect to our any study.]

4.Thank you for stating the following in the Competing Interests section:

[The authors have declared that no competing interests exist.].   

We note that one or more of the authors are employed by a commercial company: Daiichi Sankyo Co., Ltd.

Reviewers' comments:

Reviewer's Responses to Questions

**Comments to the Author**

1. Is the manuscript technically sound, and do the data support the conclusions?

Reviewer #1: Yes

Reviewer #2: Yes

2. Has the statistical analysis been performed appropriately and rigorously? 

Reviewer #1: Yes

Reviewer #2: Yes

3. Have the authors made all data underlying the findings in their manuscript fully available?

Reviewer #1: Yes

Reviewer #2: Yes

4. Is the manuscript presented in an intelligible fashion and written in standard English?

Reviewer #1: No

Reviewer #2: No

5. Review Comments to the Author

Reviewer #1: This manuscript details a significant contribution to the discovery of chemical additives which boost cell-specific protein productivity for mammalian cell cultures. However, there was substantial grammatical issues, primarily in the abstract and introduction which will require a major revision. One particular issue in the results section is the use of glucose consumption and lactate accumulation. This terminology should be clarified to be either glucose concentration, cell-specific glucose uptake rate, or cumulative glucose consumption, where applicable. Additionally, the discovered chemical additives should be compared to known additives that boost specific productivity as part of the structure-activity section.

Reviewer #2: It is significant to improve monoclonal antibody production using recombinant Chinese hamster ovary cells for medication supply and medical cost reduction. In the manuscript entitled“4-(2,5-Dimethyl-1H-pyrrol-1-yl)-N-(2,5-dioxopyrrolidin-1-yl) benzamide (MPPB) improves monoclonal antibody production in a Chinese hamster ovary cell culture”, the authors Yuichi Aki et al. have discovered a kind of new chemical MPPB to improve monoclonal antibody production effectively in recombinant Chinese hamster ovary cells through scale screening. They found that it can suppress cell growth and increase specific glucose consumption during monoclonal antibody production. In addition, it also can suppress the galactosylation on a monoclonal antibody. Further, the structure-activity study revealed that 2,5-dimethylpyrrole was the most effective partial structure of MPPB on monoclonal antibody production. The premise of this paper is compelling and the approach seems reasonable. While the results seem to contribute meaningfully to monoclonal antibody production in recombinant Chinese hamster ovary cells, several aspects of this paper - especially a more complete description and discussion of the advantage and disadvantages of this new chemical - could be improved including:

1. Why does MPPB suppress cell growth and increase specific glucose consumption during monoclonal antibody production? Also, it also can suppress the galactosylation on a monoclonal antibody. Could you please explain why?

2. Could you please explain the concentration of MPPB on cells growth and monoclonal antibody production?

3. What is the difference of MPPB from any other known chemicals for monoclonal antibody production in Chinese hamster ovary cells?

4. The authors claim MPPB increases the production of ATP, which is responsible for the increase of monoclonal antibody productivity? The reviewer wonders whether ATP has the same effect on the monoclonal antibody productivity? Why?

5. How does MPPB regulate the produce of ATP in the cells?

6. The authors suggest 2,5-dimethylpyrrole was the most effective partial structure of MPPB on monoclonal antibody production. Whether can 2,5-dimethylpyrrole affect monoclonal antibody production through affecting the production of ATP? Is this mechanism from other known chemicals used for monoclonal antibody production in the cells?

6. PLOS authors have the option to publish the peer review history of their article (what does this mean?). If published, this will include your full peer review and any attached files.

Reviewer #1: **Yes: **Tim Brantley

Reviewer #2: No

---

## [Author Response · Author response to Decision Letter 0]

31 Jan 2021

Responses to Reviewer #1:

 Thank you for your advice, which was valuable as we revised our manuscript. We appreciate your suggestions regarding grammar and scientific wording. The revised document “PONE-D-20-26236_reviewer V1_authors V1” was submitted in addition to this manuscript. The file will be shared by Yong-Bin Yan, Ph.D.

Responses to Reviewer #2:

 Thank you very much for your essential questions, which we addressed to strengthen our study. The questions helped us to improve our manuscript and to focus the direction of our future research.

---

## [Decision Letter · Decision Letter 1]

17 Feb 2021

PONE-D-20-26236R1

4-(2,5-Dimethyl-1H-pyrrol-1-yl)-N-(2,5-dioxopyrrolidin-1-yl) benzamide improves monoclonal antibody production in a Chinese hamster ovary cell culture

PLOS ONE

Dear Dr. Aki,

Thank you for submitting your manuscript to PLOS ONE. After careful consideration, we feel that it has merit but does not fully meet PLOS ONE’s publication criteria as it currently stands. Therefore, we invite you to submit a revised version of the manuscript that addresses the points raised during the review process.

We look forward to receiving your revised manuscript.

Kind regards,

Yong-Bin Yan, Ph.D.

Academic Editor

PLOS ONE

Reviewers' comments:

Reviewer's Responses to Questions

**Comments to the Author**

1. If the authors have adequately addressed your comments raised in a previous round of review and you feel that this manuscript is now acceptable for publication, you may indicate that here to bypass the “Comments to the Author” section, enter your conflict of interest statement in the “Confidential to Editor” section, and submit your "Accept" recommendation.

Reviewer #1: (No Response)

Reviewer #2: All comments have been addressed

2. Is the manuscript technically sound, and do the data support the conclusions?

Reviewer #1: Yes

Reviewer #2: Yes

3. Has the statistical analysis been performed appropriately and rigorously? 

Reviewer #1: Yes

Reviewer #2: N/A

4. Have the authors made all data underlying the findings in their manuscript fully available?

Reviewer #1: Yes

Reviewer #2: Yes

5. Is the manuscript presented in an intelligible fashion and written in standard English?

Reviewer #1: Yes

Reviewer #2: Yes

6. Review Comments to the Author

Reviewer #1: The authors of this manuscript have made substantial improvements to the manuscript. I have identified a few minor grammatical issue and substance issues with the manuscript that will need to be address prior to publication. This discovery of small molecules increasing CHO specific productivity and the investigation into their MOA is an important scientific contribution and I look forward to reviewing the the next version of this manuscript.

Reviewer #2: (No Response)

7. PLOS authors have the option to publish the peer review history of their article (what does this mean?). If published, this will include your full peer review and any attached files.

Reviewer #1: No

Reviewer #2: No

---

## [Author Response · Author response to Decision Letter 1]

2 Apr 2021

Responses to Reviewer #1:

 Thank you again for your advice, which was very insightful. We sincerely considered your advice and updated our manuscript to reflect as much of your advice as possible. Some of the sentences you pointed out were very important to our manuscript. We hope our responses below acceptably address your comments and questions.

---

## [Editor Report · Decision Letter 2]

7 Apr 2021

4-(2,5-Dimethyl-1H-pyrrol-1-yl)-N-(2,5-dioxopyrrolidin-1-yl) benzamide improves monoclonal antibody production in a Chinese hamster ovary cell culture

PONE-D-20-26236R2

Dear Dr. Aki,

We’re pleased to inform you that your manuscript has been judged scientifically suitable for publication and will be formally accepted for publication once it meets all outstanding technical requirements.

Kind regards,

Yong-Bin Yan, Ph.D.

Academic Editor

PLOS ONE
---

## [Editor Report · Acceptance letter]

12 Apr 2021

PONE-D-20-26236R2 

4-(2,5-Dimethyl-1*H*-pyrrol-1-yl)-*N*-(2,5-dioxopyrrolidin-1-yl) benzamide improves monoclonal antibody production in a Chinese hamster ovary cell culture 

Dear Dr. Aki:

I'm pleased to inform you that your manuscript has been deemed suitable for publication in PLOS ONE. Congratulations! Your manuscript is now with our production department. 

Kind regards, 

on behalf of

Dr. Yong-Bin Yan 

Academic Editor

PLOS ONE